# The Encapsulation of Bioactive Plant Extracts into the Cellulose Microfiber Isolated from *G. optiva* Species for Biomedical Applications

**DOI:** 10.3390/membranes12111089

**Published:** 2022-11-02

**Authors:** Khim Prasad Panthi, Aashish Gyawali, Shiva Pandeya, Motee Lal Sharma Bhusal, Bhanu Bhakta Neupane, Arjun Prasad Tiwari, Mahesh Kumar Joshi

**Affiliations:** 1Department of Chemistry, Trichandra Multiple Campus, Tribhuvan University, Kathmandu 44613, Nepal; 2Central Department of Chemistry, Tribhuvan University, Kathmandu 44613, Nepal; 3Mechanical Engineering and Engineering Science, the University of North Carolina at Charlotte, Charlotte, NC 28223, USA

**Keywords:** agricultural waste, cellulose microfibers (CMF), antimicrobial activity, *G. optiva*, composite

## Abstract

Agricultural waste-based cellulose fibers have gained significant interest for a myriad of applications. *Grewia optiva (G. optiva),* a plant species, has been widely used for feeding animals, and the small branches’ bark is used for making rope. Herein, we have extracted cellulose fibers from the bark of *G. optiva* species via chemical treatments (including an alkaline treatment and bleaching). The gravimetric analysis revealed that the bark of *G. Optiva* contains cellulose (63.13%), hemicellulose (13.52%), lignin (15.13%), and wax (2.8%). Cellulose microfibre (CMF) has been synthesized from raw fibre via chemical treatment methods. The obtained cellulose fibers were crosslinked and employed as the matrix to encapsulate the bioactive plant extracts derived from the root of *Catharanthus roseus (C. roseus).* The microscopic images, XRD, FTIR, and antibacterial/antioxidant activity confirmed the encapsulation of natural extracts in the cellulose microfiber. The microscopic images revealed that the encapsulation of the natural extracts slightly increased the fiber’s diameter. The XRD pattern showed that the extracted cellulose microfiber had an average crystalline size of 2.53 nm with a crystalline index of 30.4% compared to the crystalline size of 2.49 nm with a crystalline index of 27.99% for the plant extract incorporated membrane. The water uptake efficiency of the synthesized membrane increased up to 250%. The antimicrobial activity of the composite (the CMF-E membrane) was studied via the zone inhibition against gram-positive and gram-negative bacteria, and the result indicated high antibacterial activity. This work highlighted *G. optiva-derived* cellulose microfiber as an optimum substrate for antimicrobial scaffolds. In addition, this paper first reports the antimicrobial/antioxidant behavior of the composite membrane of the *C. roseus* extract blended in the *G. optiva* microfiber. This work revealed the potential applications of CMF-E membranes for wound healing scaffolds.

## 1. Introduction

*G. optiva* is a multipurpose agroforestry tree providing leaf fodder, fiber, and fuel wood. This plant species is distributed in the Himalayan regions of Nepal, India, and Pakistan, usually at an altitude of between 500 and 2500 m from sea level [1]. Traditionally, *G. optiva* fibers (GOFs) are extracted with an ecofriendly water retting process which is simple to perform and causes no harm to the fiber morphology and mechanical integrity. This process involves drying the shoots of the *G. optiva* tree after its leaves are eaten by cattle, followed by immersing the shoots in water (river, tanks, or ponds) for a month. The doused shoots are then taken out and beaten on stones, the bark of each shoot is peeled out into narrow strips, washed thoroughly in running water, and the outer pulp is cleaned by continuously washing in water to obtain the inner bark. Finally, the extracted fibers are dried under a sun bath for the complete removal of moisture. Traditionally, these fibers were used as rope for tying cattle and strings for cots. However, such natural fibers are replaced with a rope of plastics due to their long-lasting nature and excellent mechanical strength. Therefore, most of the shoots, after the leaves are eaten by cattle, remain as agroforest waste. The extraction of fibers from agroforest waste and their utilization for producing eco-friendly value-added products are promising strategies.

Natural fibers extracted from plants are composed of a complex, layered structure with hollow cellulose fibrils (α-cellulose) held together with hemicelluloses, lignin, pectin, and waxes [2]. The hydrogen bond between hemicelluloses and helically wounded cellulose microfibrils (diameter: 10–30 nm, consisting of long-chain cellulose (30–100)) acts as a cementing matrix which is a primary structural component [3]. Recently, natural fibers have been utilized for manufacturing composite materials that possess remarkable mechanical, thermal, and physicochemical properties depending on the chemical composition and structural dimensions. Plant-based natural fibers have been extensively used to extract micro and nano cellulose fibers for a myriad of applications.

Cellulose fibers derived from biomasses have unique chemical structures, adaptable surface characteristics, porous and long-chain structures, low densities, high aspect ratios, high molecular weights [4], excellent mechanical properties, biodegradable properties, and are environmentally friendly in nature [5]. Therefore, such materials show great potential in the paper and board industry, adsorbent products, wastewater treatments, drug delivery applications, tissue engineering, paints, wood coating adhesives, packaging, electronics, sensors, and medical, pharmaceutical, and cosmetic products [6,7,8,9,10]. Among the various applications, the biocompatibility and large surface area of cellulose fibers have expanded their applications in biomedical fields, such as drug delivery and tissue engineering. Irrespective of the tissue type, a number of factors, such as biodegradability, biocompatibility, mechanical properties, scaffold architecture, and manufacturing technology, are important when designing or determining the suitability of a scaffold for its usage in tissue engineering.

Skin is an important organ in the human body; it can protect and maintain environmental stability in vivo. A wound is an injury or tear on the skin surface caused by physical, chemical, mechanical, and thermal damages. Wound healing is a complex and painful process. A wound dressing is a biomedical material that protects injured skin from external risk factors during skin regeneration and accelerates the healing process by preventing wounds from infection and restoring the structure and function of the skin. Plant-based cellulose micro/nanofibers resemble the collagen fibers of native tissue and provide topographical and biochemical clues for cell differentiation and cell migration, and therefore are considered the ideal scaffold for tissue regeneration, including wound dressing. Infection is frequently encountered in wounds; microorganisms drain the resources from the surrounding skin and can colonize and interfere with the wound healing of hospitalized patients. Polymer-reinforced fillers have been widely investigated in wound healing scaffolds [11,12,13]. *Catharanthus roseus (C. roseus)* is an important medicinal plant that belongs to the *Apocynaceae* family and is an erected procumbent herb found in subtropical areas [14]. The extracts from the flowers, roots, and leaves of plants are applied to cure fresh cutting wounds by the local people in rural areas of Nepal. Recent reports revealed that the plant possesses bioactive antioxidant, antibacterial, antifungal, anticancer, and antiviral compounds [15]. In addition, previous studies demonstrated that the *Catharanthus* extract had a high rate of wound contraction, significantly decreased the epithelization period, significantly increased the dry weight and hydroxyproline content of the granulation tissue, and enhanced the viability of rat dermal fibroblasts; accordingly, the extract is expected to be a potential supplement in fibroblast cultures [16,17,18]. However, in-depth studies are lacking regarding its incorporation into fibrous scaffolds for regenerative medicine.

Therefore, this study reports the fabrication and characterization of cellulose microfibers containing different concentrations of the *C. roseus* root extract for wound healing. The cellulose microfibers were extracted from the agroforestry waste GOFs, and the antimicrobial/ antioxidant natural extract was extracted from the medicinal plant, *C. roseus*. The root extract was encapsulated into the cellulose microfiber via an immersion process. The materials were characterized with Fourier transmission infrared spectroscopy (FTIR), an optical microscope, XRD, and thermal gravimetric analysis (TGA). The antibacterial potential of *C. roseus* root extract-functionalized nano/microfibers against gram-positive and gram-negative were investigated using agar diffusion methods. The wound healing properties of different concentrations of *C. roseus* in the fibers were investigated using animal models.

## 2. Methods and Materials

All the samples were collected from the Gulmi district, located in the Western hilly region of Nepal, in January 2019. The altitude of the plant collection location was 6800 feet from sea level. The washing, drying, grinding, and sieving steps were performed before further treatments. The chemical reagents used for this work were laboratory grade and used without further purification.

### 2.1. Extraction of Cellulose Microfibers

The cellulose microfibers were extracted from the bark of “patuwa” (*G. optiva*) following a previous report with slight modifications [19,20]. Briefly, the barks of patuwa were preconditioned and washed several times with distilled water and dried in an oven at 80 °C for 2 h. The wax content from natural fibers is usually removed by boiling in a polar solvent mixture, such as a benzene-ethanol azeotrope or a toluene/ethanol mixture system [21]. Herein, we used the toluene/ethanol mixture system (2:1) to avoid benzene toxicity [20]. Toluene dissolves wax and maintains fluidity at low temperatures, while ethanol dissolves wax as well as many other polar compounds [22]. The dried samples were chopped and boiled in a toluene/ethanol mixture in the ratio of 2:1 in a Soxhlet extractor to remove the wax for 6 h, dried for 24 h in an oven (70 °C), and weighed in order to determine the content of the wax in the fiber. Later, the dried dewaxed fibers were bleached using 0.7% acidified NaClO_2_ in a magnetic stirrer at 90–95 °C for 3 h to remove the lignin content from the fiber. During this process, the chemical bonds present in the chromophore were broken into small molecules that did not absorb visible light. The hemicellulose content of the sample was removed with a treatment of NaOH (4% *w/v*) at ~60 °C for 24 h in a magnetic stirrer. The alkaline hydrolysis resulted in the substantial breakdown of the lignocellulose/hemicellulose structure and the depolymerization of lignin by affecting the acetyl group in the hemicellulose and linkage of the lignin-carbohydrate ester [23]. The resulting fibers were washed with distilled water and added to the following mixture: 250 mL of 4% NaOH and 3% H_2_O_2_. This was kept in a water bath at 80 °C for an hour, followed by filtration. The obtained fibrous mass was added to 1% H_2_O_2_ (250 mL), stirred for an hour at 85 °C, and washed with distilled water until the pH turned neutral. The material was then allowed to air dry and be weighed.

### 2.2. Extraction of C. roseus Roots

The roots of *C. roseus* were ground in an electric blender to create a fine powder. The powder was packed in filter paper. The apparatus was set for a Soxhlet extraction with the wrapped sample of root powder (40 g). The Soxhlet extraction process was carried out for 60 min. Here, ethanol was used as a solvent. The obtained extract was evaporated in a water bath at 45 °C for several days. It was then stored in a cold place and used for further study.

### 2.3. Preparation of the Plant Extract Incorporated Membrane (CMF-E)

The CMF-E membrane was prepared through the adsorption of the ethanolic extract of *C. roseus* onto the surface of the cellulose fibrils following previous reports with slight modifications [24,25]. Briefly, the cellulose microfiber suspension was dispersed into deionized water with aliquots of an aqueous glycerol solution (3%). A specific volume of the extract solution (100 mg/mL) was then slowly added dropwise to the cellulose microfiber suspension and magnetically stirred for 30 min. The suspension was then cast into a 90 mm diameter dish and dried at room temperature for 48 h. The cellulose membrane without the plant extract (CMF) was prepared for comparison following a similar procedure.

### 2.4. Measurement of Apparent Densities

The apparent densities of the CMF membranes were measured as reported elsewhere [25]. Briefly, the weight, thickness, and grammage (the base weight in grams per square meter) of the prepared membranes were carefully weighted, and the apparent densities of the CMF membranes, ρ, were calculated using the following equation.
ρ = (weight of the CNF membrane)/(area × thickness)(1)

### 2.5. Chemical Composition Measurement of GOFs Fibers

The chemical composition of the GOFs fibers was measured using the gravimetric method. The method involved a chemical treatment followed by weight measurement and calculating the weight loss.

### 2.6. Physicochemical Characterization

The surface morphology of the synthesized cellulose microfibers and CMF-E membranes were studied using bright field microscopy (Amscope, Irvine, CA, USA). The Fourier transform infrared (FT-IR) spectra of different samples were recorded using a spectrometer (IR Tracer 100, Kyoto, Japan). Information about the phase and crystallinity was obtained with a Rigaku X-ray diffractometer (Rigaku, Tokyo, Japan) with Cu Kα (λ = 1.540 Å) radiation over Bragg angles ranging from 10° to 80°. The crystal size and crystallinity index (CI) were evaluated from the XRD pattern by using the Scherrer equation and the origin pro 2016 (64 bit) software (OriginLab, Northampton, MA, USA), respectively.

### 2.7. Antibacterial Activity

The zone inhibition method was carried out using *Enterococcus faecalis (E. faecalis)*, *Staphylococcus aureus* (*S. aureus), Staphylococcus epidermidis (S. epidermidis), and Baccilus subtilis (B. subtilis)* as the model organisms to measure the antibacterial activity of the synthesized membranes at room conditions. Using a spread plate method, nutrient agar plates were incubated with 1 mL of bacterial suspension containing approximately 10^6^ colony forming units (CFU)/mL. The membrane samples were prepared by cutting them down into pieces with a diameter of 10 mm. Samples with the same size (a 10 mm diameter) were placed on the inoculated plates and were then incubated at 37 °C for 24 h. The zones of inhibition were determined by measuring the clear area formed around each sample.

### 2.8. Evaluation of Plant Extract Encapsulation in CMF Membrane

The amount of plant extract encapsulated in the cellulose fibers was determined spectrophotometrically. A calibration curve was established from the known concentrations of the root extract in methanol. The samples were immersed and shaken in a root extract solution of a known concentration (400 µg/mL) for 24 h. Later, the sample was removed, and the concentration of the remaining solution was determined spectrophotometrically.
Concentration of sample= (Absorbance of sample − intercept))/slope(2)

In the gravimetric analysis, the different weights of the CMF membrane before and after the extract encapsulation were determined.

### 2.9. Water Uptake of CMF-E Membrane

The water uptake of the membrane was based on the total immersion technique. In this method, the CMF-E membrane (an area of 4 cm^2^ and a thickness of 0.162 cm) was immersed in deionized water and weighed. The water uptake was measured at different time intervals, t, from 1 to 10 min. After immersion, the excess water was removed using tissue paper, and the material was weighed again. The water uptake was calculated with the following formula [25].
Water uptake = (m (t) − m (before)) × 100(3)
where m (before) and m (t) is the weight before the immersion and at the different intervals of time, respectively.

### 2.10. Active Principle Ingredients (API) (Extract) Release from CMF-E

The release of the root extract from the CMF-E membrane on the wound environment was studied by immersing the CMF-E membrane in water over different time intervals of up to 10 min. At every 1-min interval, 3 mL of a solution was taken, and the extract release as a drug was studied on the free radical scavenging activity (antioxidant properties) of a release extract using the DPPH assay method because plants extracts act as an essential source of natural antioxidants in the wound healing process.

## 3. Results and Discussion

### 3.1. Physiochemical Characterization of G. Optiva Fibers

The chemical composition, apparent density, and porosity of the *G. optiva* fibres were characterized with standard methods available in the literature, and the obtained result has been described in the details as follows. The *G. optiva* fibers (GOFs) were extracted via the traditional water retting process. Figure 1 shows the schematic illustration for the extraction of the cellulose microfibers from the GOFs. The samples were evaluated to determine the lignin and hemicellulose content of the fiber, which was found to be 15.13 Wt% and 13.52 Wt%, respectively. Furthermore, the samples were treated with H_2_O_2_ to remove the trace amounts of lignin, and the yield of the extracted cellulose from the fibers was determined. Figure 2 shows the chemical composition of the GOFs determined with gravimetric analysis using selective chemical treatments. The results revealed that cellulose was the major component of the fiber (63.13%), followed by lignin (≈15.13%), hemicellulose (≈13.52%), and wax (≈2.8%). We have compared the chemical composition of the GOFs with some important agroforestry wastes and presented this in Table 1. These results showed that jute fibers and GOFs have a very high cellulose content compared to the hardwood stem, softwood stem, and Ficus leaves.

The physical properties of the cellulose microfibrous mesh were studied in terms of the grammage, thickness, and apparent density, and are summarized in Table 2. The thickness of the CMF membranes was measured with a screw gauge micrometer, while the grammage and apparent density were calculated using a method reported elsewhere. The CMF1 sample showed a 0.030 g/cm^2^ grammage and a 0.047 cm thickness with an apparent density of 0.638 g/cm^3^, and the CMF2 sample showed a 0.026 g/cm^2^ grammage, 0.026 cm thickness, and an apparent density of 0.486 g/cm^3^. Similarly, sample CMF3 showed a thickness of 0.062 cm, a grammage of 0.032 g/cm^2^, and an apparent density of 0.062 g/cm^3^. The apparent average density of the CMF mesh was found to be 0.548 g/cm^3^, which is less than the theoretical density of cellulose (1.5 g/cm^3^) [22], suggesting that the prepared membranes were highly porous and suitable for tissue engineering applications, particularly regarding the absorption of exudate as well as cell proliferation and differentiation. From the average apparent density measured, the porosity of the CMF membrane was found to be 63.46%, calculated by using the following equation: ε (%) = (1 − ρ m /ρ cellulose × 100), where ρ_cellulose_ is the density of the cellulose (1.5 g cm^−3^) [29].

In this study, we intended to develop a bioactive antimicrobial wound dressing. Therefore, we incorporated the *C roseus* root extract into the CMF fibrous mesh to impart antibacterial activity via an immersion process. *C. roseus* is an important medicinal plant, and the extracts from its flowers, roots, and leaves possess bioactive antioxidant, antibacterial, antifungal, anticancer, and antiviral compounds. The composite *C. roseus* root extract encapsulated cellulose micro/nanofiber mesh (CMF-E) was characterized using various techniques and studied for wound healing applications.

### 3.2. X-ray Diffraction (XRD) Analysis

The XRD patterns of the different samples were studied using an X-ray diffractometer to evaluate the crystalline morphology and different phases of the materials. Figure 3 shows the X-ray diffraction patterns of the cellulose microfibers and CMF-E membranes. The CMF shows well-defined diffraction peaks at 15.77, 16.87, and 22.99, corresponding to the cellulose Iβ crystal planes (110), (110), and (200), respectively, which are in good agreement with previous reports [30,31]. The CMF-E membrane showed diffraction peaks at 15.46, 17.57, 23.34, and 35.61, representing the cellulose Iβ crystal planes (1-10), (110), (200), and (040), respectively [32]. The peak intensity of the plant extract encapsulated composite fiber matrix was slightly decreased compared to the pristine cellulose matrix (Figure 3), indicating a reduction in the degree of crystallinity of the cellulose after the encapsulation of the plant extract. The lower peak intensity of the CMF-E membrane compared to that of the CMF is attributed to the superposition of all the constructive and destructive contributions guided by the absorbed plant extract in the CMF-E membrane [33]. Further mechanical stirring during the encapsulation of the CMF-E membrane might have deformed the cellulose crystal, leading to broad diffraction peaks [30]. The XRD diffraction patterns were further employed to determine the following parameters; full width at half maximum (FWHM), d-spacing, crystal size (L), and crystallinity index (CI), and the results are shown in Table 3. The average crystalline size for the different samples was calculated using the Scherer equation and found to be 2.53 nm, and 2.49 nm for the CMF and CMF-E membranes, respectively, and the crystallinity index was determined to be 30.04% and 27.99% for the CMF and CMF-E membranes, respectively. This result revealed that the encapsulation of the extract into the cellulose chain affected its physical parameters, including the crystalline size and crystallinity of the composite fibers. The change in the peaks’ positions, the peak intensity, and the peak width confirmed the adsorption of the extract on the CMF.

### 3.3. Fourier Transform Infrared (FTIR) Spectroscopy Analysis

FTIR spectroscopy was applied to identify the potential functional groups present in the CMF and CMF-E membranes. The peaks in the range of 3660–2900 cm^−1^ are attributed to the characteristic of the stretching vibration of the O-H and C-H bonds in polysaccharides (Figure 4). The broad peak observed around 3345 cm^−1^ is attributed to the stretching vibration of the hydroxyl group in cellulose, including the inter and intramolecular hydrogen bond vibrations in cellulose [34]. The absorption bands at appoximately 2872 cm^−1^ indicated the C–H stretching vibrations of all of the hydrocarbon constituents of cellulose [35]. The peak around 1642 cm^−1^ indicates the vibration of the water molecules absorbed in cellulose [28]. The absorption bands at 1420, 1358, 1335, 1023, and 880 cm^−1^ are associated with the stretching and bending vibrations of -CH_2_ and CH, -OH, and C-O-C bonds in cellulose [28,29]. The band at 880 cm^−1^ is attributed to the C-O-C stretching vibration of the (1 → 4) glycosidic linkages between the glucose units in cellulose (the amorphous region) [34,36]. The band around 658 cm^−1^ is associated with the O-H out-of-plane bending [33]. The absence of the bands between 1500 and 1600 cm^−1^ corresponds to the lignin band, and the band at 1730–1740 cm^−1^ corresponds to hemicellulose [37] and reveals that all the lignin and hemicellulose content of the GOFs was completely removed after the chemical treatment and the cellulose microfiber obtained was highly pure with a negligible number of impurities. Furthermore, the plant extract encapsulated sample showed additional peaks at 2930 cm^−1^, 2644 cm^−1^, and 1392 cm^−1^. The results revealed that pure cellulose microfibers were extracted from the plant fiber, and the plant extract was well encapsulated in the CMF-E membrane.

### 3.4. Microscopic Characterization

Bright field microscopy is the most straightforward and popular optical microscope technique. It provides information about the samples’ surface structure and morphology. Figure 5 shows the microscopic images and corresponding histograms of the average fiber size distribution of the different samples. The microscopic images revealed that the fibers were distinctly separated from each other, with few deformities (highlighted in yellow), while some lamellar fibers were detached from the cell wall (highlighted in red) [38]. The overall morphology of the fibers remained undamaged; however some portions of the microfibers were partially damaged, which is attributed to the cellulose chain breaking at the amorphous regions [38]. The average fiber diameters of the pristine and plant extract incorporated samples were determined in order to study the effect of the extract on the fibers’ diameter distribution. The CMF membrane showed an average diameter of 4.785 ± 0.588 µm, while the CMF-E membrane had an average diameter of 7.627 ± 0.644, suggesting that the incorporation of the plant extract slightly increased the fibers’ diameter. From these findings, it can be concluded that the extract was successfully incorporated in the CMF, as the average width of the fiber increased slightly in the CMF-E membrane compared to the CMF membrane.

### 3.5. Water Uptake Efficiency

The water uptake efficiency of wound dressing materials is crucial for the absorption of bodily fluid, transfer of nutrients, cell suspension distribution, and the structural morphology of regenerated tissue [39]. Herein, we immersed the scaffold in deionized water (for different time intervals), the excess surface water was removed with filter paper, and the water uptake efficiency was determined as described elsewhere. The water uptake for the CMF and CMF-E membranes quickly reached the maximum (250%) in the first two minutes, and after that, it appeared to be almost constant, as shown in Figure 6. Begum et al. reported similar reports for different alkali-treated natural fibers [40]. Water is a strong polar compound and is easily attracted by the polar -OH groups of all cellulosic fibers [40]. The high capillary absorption of the CMF membrane is attributed to the hydrophilicity of the cellulose caused by the hydroxyl group present in the cellulose chain, as discussed in a previous study [41]. In the wound healing application, such scaffold must tend to uptake water and retain it for a long time. In this study, an initial measurement revealed the high performance of the membrane toward water uptake. In the available literature, it has been found that water uptake for cellulose-derived micro/nano fibres has been measured for hours and a gradual increase in the water uptake percentage occurs with time [42,43].

### 3.6. Antimicrobial Screening of Different Membranes

Bacterial infection is the prime concern during wound care management, and preventing bacterial infection is crucial in the development of wound dressing materials. The effectiveness of wound dressings against bacterial infections can be evaluated by observing the zone of inhibition using gram-negative and gram-positive model bacteria [44]. Herein, the inhibitory effect of the pristine CMF and CMF-E scaffold was evaluated by observing the zones of inhibition against gram-positive bacterial strains that are found primarily on wounds *(S. aureus, S. epidermidis, E. faecalis,* and *B. subtilis*). In order to perform the antimicrobial test for the selected bacterial species, the sample was prepared with an average diameter of 10 mm. Previous studies reported that gram-positive bacterial strains are found more frequently on wounds (92%) compared with gram-negative strains (7%) [45,46]. Figure 7 shows the zones of inhibition shown with the CMF and CMF-E membranes against the different bacterial strains. For the CMF-E membrane, the zone inhibition against *S. aureus*, *S. epidermidis*, *E. faecalis* and *B. subtilis* was found to be 12.30 mm, 9.82 mm, 8.80 mm, and 10.10, respectively, while the pristine CMF membrane showed no effect against any bacterial strain. This result revealed that the natural extract was well incorporated into the cellulose microfibers, and the composite CMF-E membrane showed strong antibacterial activity against different bacterial strains. The plant extract may have strong hydrogen bonding interactions or covalent interactions with the hydroxyl groups of the cellulose chain present in the CMF membrane.

## 4. Conclusions

Micro/nanocellulose fibers (NFCs) were successfully isolated using eco-friendly, rapid, simple, and low-cost top-down approaches from locally produced “patuwa”, the bark of *Grewia optiva*. The FTIR analysis further confirmed the successful fabrication of pure nanocellulose fibers, as indicated by the presence of the infrared spectra corresponding to the cellulose fibers. The CNF membranes with the incorporated extract (CNF-E) were fabricated by an immersion method and were characterized with XRD and bright field microscopy techniques. The bright field microscope images showed tapered-shaped CNF fibers, which were 50 µm wide and were not altered after antibacterial *C. roseus* extract entrapment. The XRD confirmed the formation of cellulose with an average crystalline size of 2.3 nm for the CNF and 1.8 nm for the CNF-E membrane. When comparing the two XRD peaks, the peak intensity of the CNF-E membrane was found to be lower than the CNF; this is due to the superposition of all the constructive and destructive contributions, which is guided by the position of the absorbed extract atoms to the CNF. The theoretical density of the CNF-E membranes was 1.43 g/cm^3^, which is close to the theoretical density of cellulose (1.5 g/cm^3^). The antimicrobial screening showed vigorous antibacterial activity against gram-positive bacterial strains (*S. epidermidis, E. faecalis, B. subtills,* and *S. aureus*) commonly found in the wounded environment. The water uptake study suggests a high water uptake capacity for CNF membranes (around 250%), reinforcing the interest in CNF for wound applications. From the above findings, it is concluded that the green product comprised of an extraction of *C. roseus* encapsulated with a CNF membrane could be a good choice for future study.

## Figures and Tables

**Figure 1 membranes-12-01089-f001:**
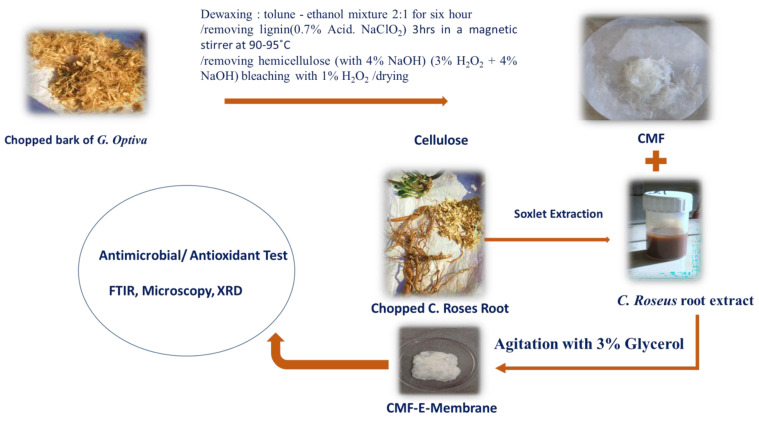
Schematic illustration showing the extraction of cellulose microfibers (CMF) from *G. optiva* fibres GOFs and encapsulation of *C. Roses* root extract to synthesise a cellululose micro fibre membrane (CMF-E).

**Figure 2 membranes-12-01089-f002:**
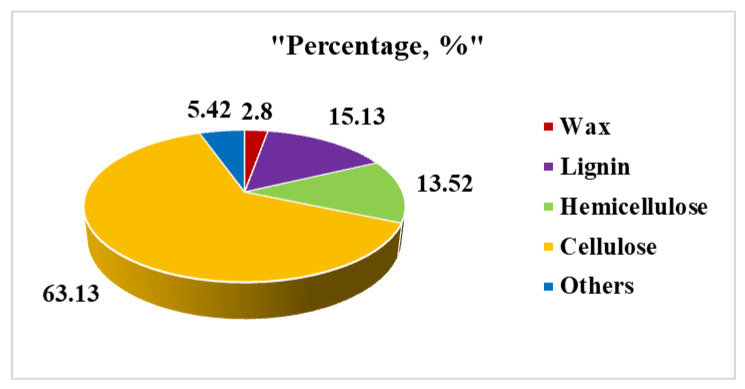
Pie chart showing the chemical composition of GOFs fibers via a gravimetric analysis.

**Figure 3 membranes-12-01089-f003:**
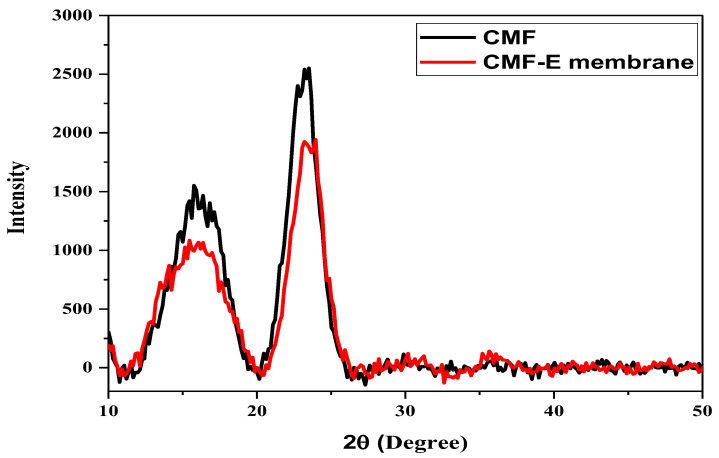
XRD patterns of the different samples.

**Figure 4 membranes-12-01089-f004:**
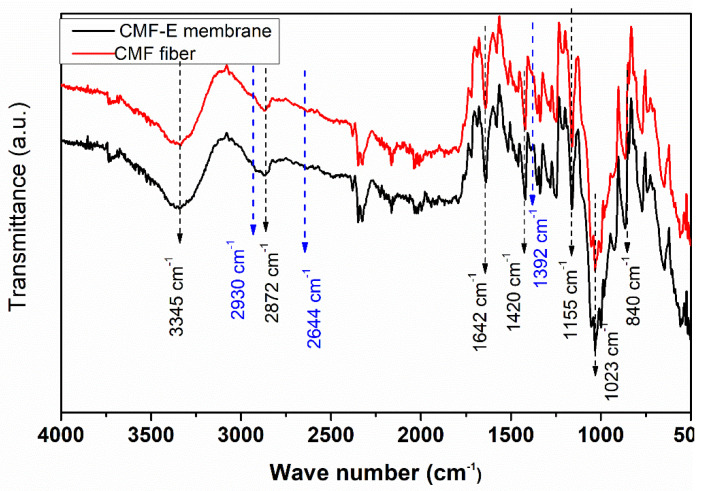
FTIR spectra of the different samples.

**Figure 5 membranes-12-01089-f005:**
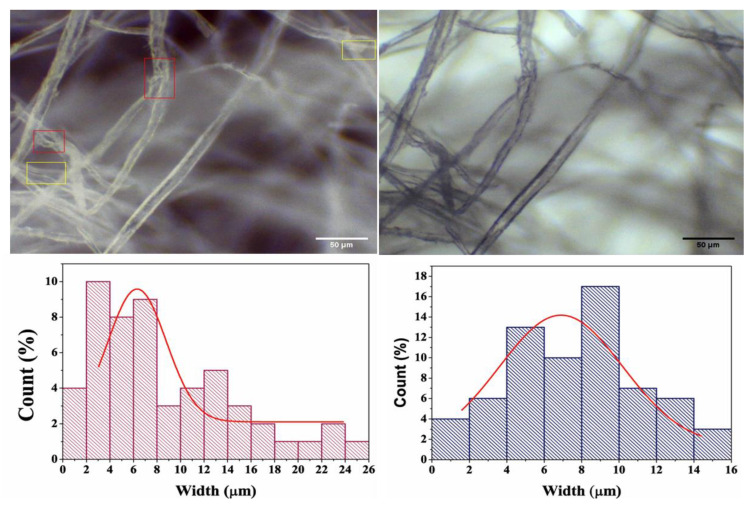
Microscopic images of the different samples: Cellulose microfiber (CMF) (left), and cellulode microfiber plant extract encapsulated (CMF-E) membrane (right), and the corresponding fiber size distributions. The red and yellow selected areas in top left figure are the smallest fiber regions.

**Figure 6 membranes-12-01089-f006:**
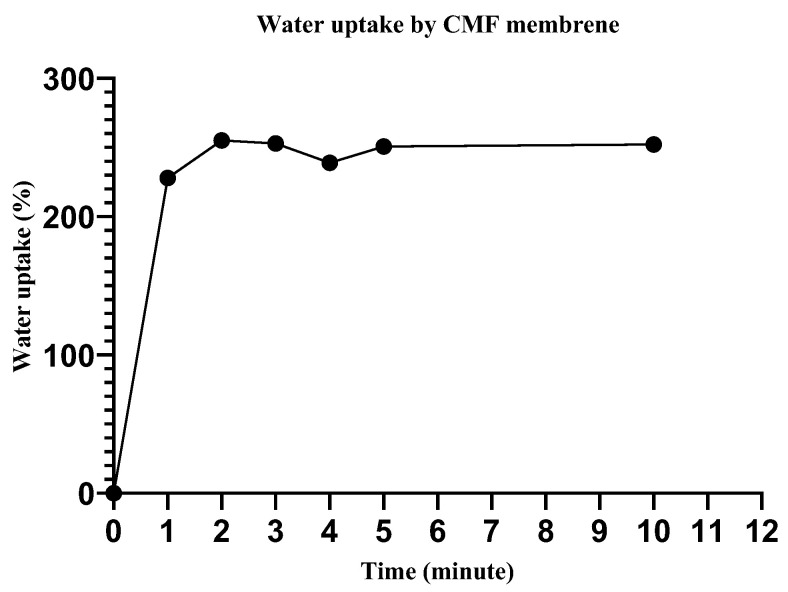
Water uptake efficiency by the cellulose microfiber (CMF) membrane.

**Figure 7 membranes-12-01089-f007:**
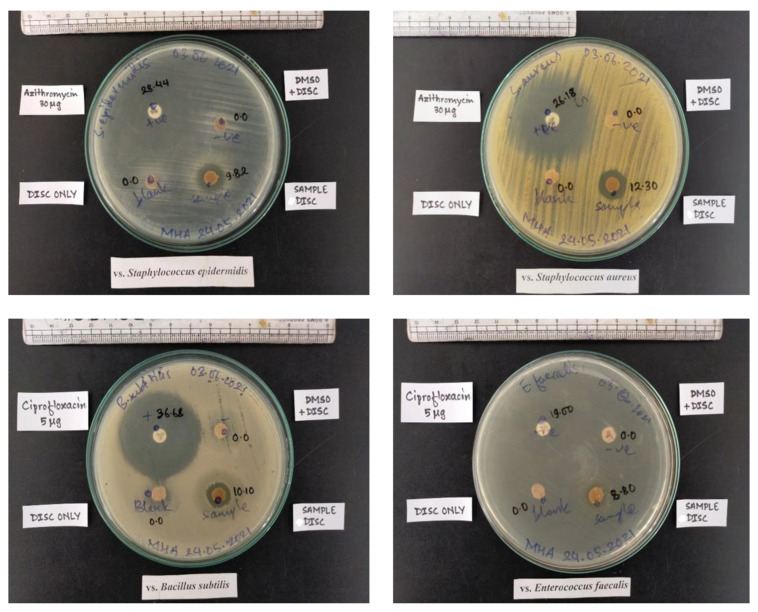
Antibacterial activity of different samples against different bacterial strains.

**Table 1 membranes-12-01089-t001:** Comparison of the chemical composition of some important agroforestry waste.

Agroforestry Waste	Cellulose (%)	Hemicellulose (%)	Lignin (%)	Reference
Hardwood stem	40–45	24–40	18–25	[26]
Softwood stem	45–50	25–30	25–35	[26]
Ficus leaves	38.1	30.5	23.4	[27]
Jute	61–71	14–20	12–13	[3,28]
*G. optiva* (bark)	≈63.3	≈13.52	≈15.13	Present work

**Table 2 membranes-12-01089-t002:** Physical properties of the CMF membranes.

Area (cm^2^)	Weight (g)	Thickness (cm)	Grammage (g/cm^2^)	Apparent Density (g/cm^3^)
CMF1	0.030	0.047	0.030	0.638
CMF2	0.058	0.053	0.026	0.486
CMF3	0.129	0.062	0.032	0.520

**Table 3 membranes-12-01089-t003:** XRD results of the cellulose microfiber (CMF) and CMF-E membrane.

Sample	Peak Position (2θ) in Degrees	d-Spacing (A) d_200_	FWHM (Degree)	Crystalline Size L (nm)	Crystallinity Index
CMF	22.99	3.86	3.17	2.53	30.04
CMF-E	23.34	3.80	3.22	2.49	27.99

Note: The systematic error correction for 2θ was (−0.5) degrees.

## Data Availability

Data sharing not applicable as all the data are included in the manuscript. No new data were created or analyzed in this study. Data sharing is not applicable to this article.

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
