# Peer review of "The Encapsulation of Bioactive Plant Extracts into the Cellulose Microfiber Isolated from G. optiva Species for Biomedical Applications"

_membranes, 2022, doi:10.3390/membranes12111089_

Round 1
Reviewer 1 Report
This manuscript entitled "Encapsulation of bioactive plant extract into the cellulose micro/nanofiber isolated from Grewia optiva species for biomedi- cal application" by Panthi et al. is interesting and suitable to this journal. However, after I go through, some improvements must be done before it can be considered for publication. My suggestions are as follow:
1. Why did authors treat the fiber with acid hydrolysis?. Acid hydrolysis is usually use to produce cellulose nanocrystals. However, author did not produce it. They produce cellulose microfibers instead of cellulose microcrystals. Please justify, why they did acid hydrolysis treatment?. Otherwise, just remove it.
2. Authors cannot use the "nano" term. Based on the TEM image shown in Fig. 4, the diameter size of the treated fiber is very larger (estimated more than 100nm). So it should be classified as cellulose fibers. Authors must use the term "cellulose fiber" instead.
3. Title: Must be improved. Authors must include "cellulose microfiber". In this case, I dont think acid hydrolysis treatment gives any significant effect to their study.
4. Carefull with abbreviation. What the CMF, and CNF?. Please check through out the manuscript.
5. Figure 1 and 3 must be improved especially on formatting. The Figure 1 image is low. Please check and improve.
6. Grewia optiva or G. Optiva?. Please standardize.
7. Please updated recent papers in the literature review to give more sense to the reader. Some improtant statements need to have references especially to these sentences:
"This plant species is distributed in Himalayan regions of Nepal, India 34 and Pakistan usually between 500 and 2500 m altitude from sea level."
"The hydrogen bond between hemicelluloses and helically wounded cellulose 51 microfibrils (diameter: 10-30 nm, consists of long-chain cellulose (30-100) acts as a cement- 52 ing matrix which is a primary structural component."
These are some related paper to be cited in the revised manucript.
a. Structural Strength Analyses for Low Brass Filler Biomaterial with Anti-Trauma Effects in Articular Cartilage Scaffold Design. Materials (Basel). 2022;15(13):4446. doi:10.3390/ma15134446
b. Structural Characterization Analyses of Low Brass Filler Biomaterial for Hard Tissue Implanted Scaffold Applications. Materials (Basel). 2022;15(4):1421. doi:10.3390/ma15041421.
c. Numerical Simulation Study on Relationship between the Fracture Mechanisms and Residual Membrane Stresses of Metallic Material. J Funct Biomater. 2022;13(1):20. doi:10.3390/jfb13010020
8. Please check the citation formatting. [1] or (John & Anandjiwala, 2008)
9. What is the different between left and right image of Figure 4. No indication.
10. In the method section, authors did mention they the antibacterial test was performed using Enterococcus faecalis. But, in the result, they mentioned 4 species which are S. aureus, S. epidermidis, E. faecalis, B. subtilis. Please careful describe and make it clearly.
Author Response
Reviewer 1
This manuscript entitled "Encapsulation of bioactive plant extract into the cellulose micro/nanofiber isolated from Grewia optiva species for biomedi- cal application" by Panthi et al. is interesting and suitable to this journal. However, after I go through, some improvements must be done before it can be considered for publication. My suggestions are as follow:
Our response: Thank you very much for your kind remarks to improve the manuscript. Revisions were highlighted in green color.
- Why did authors treat the fiber with acid hydrolysis?. Acid hydrolysis is usually use to produce cellulose nanocrystals. However, author did not produce it. They produce cellulose microfibers instead of cellulose microcrystals. Please justify, why they did acid hydrolysis treatment?. Otherwise, just remove it.
Our response: Thank you very much for your kind suggestions. We have revised the schematic outline and replaced in revised manuscript. (page number 5, result and discussion section). The acid hydrolysis part has been removed in revised manuscript.
- Authors cannot use the "nano" term. Based on the TEM image shown in Fig. 4, the diameter size of the treated fiber is very larger (estimated more than 100nm). So it should be classified as cellulose fibers. Authors must use the term "cellulose fiber" instead.
Our response: Thank you very much for your kind suggestion. We have corrected accordingly and used the term cellulose micro fiber throughout the revised manuscript including a title (page 1, title).
- Title: Must be improved. Authors must include "cellulose microfiber". In this case, I dont think acid hydrolysis treatment gives any significant effect to their study.
Our response: Thank you for your kind remarks. We have corrected the title in revised manuscript accordingly. We have changed the schematic figure removing the acid hydrolysis part (page number 5, figure 1)
- Carefull with abbreviation. What the CMF, and CNF?. Please check throughout the manuscript.
Our response: Thank you for your minute correction. We have used the abbreviation CMF throughout uniformity has been maintained throughout by using the term CMF including in the XRD/FTIR plot (page number 7, figure 3).
- Figure 1 and 3 must be improved especially on formatting. The Figure 1 image is low. Please check and improve.
Our response: Thank you for your kind suggestion. We have improved the resolution of Figures in the revised manuscript. New XRD and FTIR plot has been included (page number 7, figure 3 and page number 5, figure 1)
- Grewia optiva or G. Optiva?. Please standardize.
Our response: Thank you for your minute correction. We have used the term G. optiva throughout the revised manuscript maintain the uniformity.
- Please updated recent papers in the literature review to give more sense to the reader. Some improtant statements need to have references especially to these sentences:
"This plant species is distributed in Himalayan regions of Nepal, India 34 and Pakistan usually between 500 and 2500 m altitude from sea level."
Our response: Thank you very much for your kind suggestion. We have added the new reference accordingly. (Ref 1, first page, first paragraph.)
"The hydrogen bond between hemicelluloses and helically wounded cellulose 51 microfibrils (diameter: 10-30 nm, consists of long-chain cellulose (30-100) acts as a cement- 52 ing matrix which is a primary structural component."
Our response: Thank you very much for your kind suggestion. Due citation has been included in the revised manuscript ( Ref. No. 3, page 2, second paragraph)
These are some related paper to be cited in the revised manucript.
- Structural Strength Analyses for Low Brass Filler Biomaterial with Anti-Trauma Effects in Articular Cartilage Scaffold Design. Materials (Basel). 2022;15(13):4446. doi:10.3390/ma15134446
Our response: Thank you very much for kind suggestion. We have added the aforementioned reference in revised manuscript (Ref. No. 11, page – 2)
- Structural Characterization Analyses of Low Brass Filler Biomaterial for Hard Tissue Implanted Scaffold Applications. Materials (Basel). 2022;15(4):1421. doi:10.3390/ma15041421.
Our response: Thank you very much for kind suggestion. We have included the aforementioned reference in revised manuscript (Ref. No. 12 , page – 2)
- Numerical Simulation Study on Relationship between the Fracture Mechanisms and Residual Membrane Stresses of Metallic Material. J Funct Biomater. 2022;13(1):20. doi:10.3390/jfb13010020
Our response: Thank you very much for kind suggestion. We have included the aforementioned reference in revised manuscript (Ref. No.13 , page – 2)
- Please check the citation formatting. [1] or (John & Anandjiwala, 2008)
Our response: Thank you very much for your minute correction. We have corrected citation formatting throughout in the revised manuscript ( page number 2, second paragraph)
- What is the different between left and right image of Figure 4. No indication.
Our response: Thank you for your quires. The caption of Fig 4. Has been revised accordingly (figue 4, page number 8).
- In the method section, authors did mention they the antibacterial test was performed using Enterococcus faecalis. But, in the result, they mentioned 4 species which are S. aureus, S. epidermidis, E. faecalis, B. subtilis. Please careful describe and make it clearly.
Our response: Thank you very much for your minute correction. The antibacterial test was performed for all four aforementioned mentioned bacteria. We have described clearly in method and materials section of revised manuscript (page number 2, antibacterial test section)

Reviewer 2 Report
In this manuscript, the authors reported the extraction of cellulose micr/nano fibres from Grewia optiva and used it for the encapsulation of C. roseus roots extract to prepare potential wound dressing. I strongly find this investigation of high interest, but I have some major concerns that made me recommend “Major revisions”.
· Abstract: In this section it is not clear the aim of the study. I mean which are the benefits of those bioactive extracts, which are those extract and their target. That should be briefly incorporated in this section. Indeed, although the general idea of what is done in this resarch can be easily seen, its novelty seems not to have been suitably highlighted.
o Line 14: “Gravimetric analysis”…..In addition, the thermal character of the TGA result should be also highlighted, instead oof just “Gravimetric
· Introduction:
o The authors used a changing referenc format (lines 51 and 55 for exampple). Please, unify it in the whole manuscript.
· Materials and Methods.
o Line 107: “…Gulmi district of Nepal in January 2019.” I would recommend the authors to extent a little bit origin of the sample, since that seems to be quite general.
o Lines 117-120: Are the removal of lignin and hemicellulose by means of their solubilization and a subsequent filtration process? In that case (and if not as well) please complete this methodological part.
o Lines 140-144: Which were tha sample shapes?
o Lines 170-177: Why did the authors select 10 min as the maximum water uptake time? Why not longer or shorter? Is the potential final product only applicable for that time span in an animal body?
o Lines 188-203: All these lines seem to belong to the experimental section. I would recommend to summarize this part and complete that one located h¡in the experimental section.
o Line 221: It seems there is one missing reference.
o Why did not the authors evaluate the porosity of these systems? That appears to be fundamental when dealing with these kind of materials and structures (even more importantly when considering their ultimate application target).
o The figure plotting the FTIR spectra is required.
o Line 293: “…(highlighted in green)” There is nothing highlighted in green. Please check it and rephrase it.
o Lines 185-299: The authors are encouraged to include a title (3.X…) of the first result section discussion.
o In figure 4 it is required to indicate which of these diagramas correspond to either CMF or CMF-E.
o Lines 351-352: “…extraction of C. roseus encapsulated CNF membrane can be a good choice for faster wound healing” Have the authors compared the velocity wound healing of their design dressing wound materials with some references? If not, this statement remains unfounded.
Author Response
Reviewer 2
Abstract:
In this section it is not clear the aim of the study. I mean which are the benefits of those bioactive extracts, which are those extract and their target. That should be briefly incorporated in this section. Indeed, although the general idea of what is done in this research can be easily seen, its novelty seems not to have been suitably highlighted.
Our response: Thank you very much kind suggestion to enhance the quality of our manuscript. We have elaborated the abstract and mentioned the novelty of our present work. (last sentences of abstract page)
- Line 14: “Gravimetric analysis”…..In addition, the thermal character of the TGA result should be also highlighted, instead oof just “Gravimetric
Our response: Thank you very much for your kind suggestions. Herein we have performed the gravimetric analysis to quantify the component of raw fiber. We agree with reviewer that the TGA result would be more advantageous. However, due to the problem with our TGA instrument, we are unable to include the TGA graph. We will include the TGA data for upcoming manuscript. We hope you will consider situations.
- Introduction:
- The authors used a changing reference format (lines 51 and 55 for example). Please, unify it in the whole manuscript.
Our response: Thank you very much for minute correction. We have corrected the citation formatting throughout the text in revised manuscript (e.g page number 2, reference number 3).
- Materials and Methods.
- Line 107: “…Gulmi district of Nepal in January 2019.” I would recommend the authors to extent a little bit origin of the sample, since that seems to be quite general.
Our response: Thank you very much for your kind suggestion. The description has been elaborated regarding the plant location in revised manuscript. (Page number 3, materials and methods part, and first paragraph)
- Lines 117-120: Are the removal of lignin and hemicellulose by means of their solubilization and a subsequent filtration process? In that case (and if not as well) please complete this methodological part.
Our response: Thank you very much for your kind suggestion. We have added the following text in the e methodological part of revised manuscript (page number 3).
The dried samples were chopped and boiled in a toluene/ethanol mixture in the ratio of 2:1 in a Soxhlet to removed wax for 6 hour and dried for 24h in oven (70oC) to obtain nonwoven fibers. Later, the dried dewaxed fibers were bleached using 0.7% acidified NaClO2 in a magnetic stirrer at 90-95 ËšC for 3 h to remove of lignin content from the fiber. The hemicelluloses content of the sample was removed by treating with NaOH (4% w/v) at ~60 ËšC for 24 h in a magnetic stirrer. The resulting fibers were washed with distilled water and added to the mixture 250 mL of 4% NaOH and 3% H2O2 kept in a water bath at 80 ËšC for an hour followed by filtration. The obtained fibrous mass was added to 1% H2O2 (250 mL) and stirred for an hour at 85ËšC and washed with distilled water until the pH turned neutral and was allowed to air dry and weighed.
- Lines 140-144: Which were the sample shapes?
Our response: Thank you very much for your kind concern. The samples were of 10 mm in diameter. We have added in revised manuscript (page number 9, antimicrobial test chapter)
- Lines 170-177: Why did the authors select 10 min as the maximum water uptake time? Why not longer or shorter? Is the potential final product only applicable for that time span in an animal body?
Our response: Thank you very much for your kind concern. The water uptake efficiency of as-synthesized material remained constant after 10 min. Therefore we have showed the data for 10 min.
- Lines 188-203: All these lines seem to belong to the experimental section. I would recommend to summarize this part and complete that one located h¡in the experimental section.
Our response: Thank you very much for your kind suggestion. We have moved this part in experimental section in revised section.
- Line 221: It seems there is one missing reference.
Our response: Thank you very much for your minute correction. We have added the reference in revised manuscript.
- Why did not the authors evaluate the porosity of these systems? That appears to be fundamental when dealing with these kind of materials and structures (even more importantly when considering their ultimate application target).
Our response: Dear reviewer, porosity has been calculated by apparent density (page number 6, last paragraph)
The figure plotting the FTIR spectra is required.
Our response: Thank you very much for your kind concern. FTIR plot has been in revised manuscript.
- Line 293: “…(highlighted in green)” There is nothing highlighted in green. Please check it and rephrase it.
Our response: Thank you very much for your minute correction. We have corrected accordingly in revised manuscript.
- Lines 185-299: The authors are encouraged to include a title (3.X…) of the first result section discussion.
Our response: Thank you very much for minute correction. We have revised accordingly ( 3.1 Physio-chemical characterization of G. optiva fiber, page number 5, 3.2 XRD, FTIR page number 7, 3.3 microscopic characterisation page number 8)
- In figure 4 it is required to indicate which of these diagramas correspond to either CMF or CMF-E.
Our response: Thank you very much for your kind suggestions. We have revised Fig. 4 accordingly and included in revised manuscript page number 9)
o Lines 351-352: “…extraction of C. roseus encapsulated CNF membrane can be a good choice for faster wound healing” Have the authors compared the velocity wound healing of their design dressing wound materials with some references? If not, this statement remains unfounded.
Our response: Thank you very much for your kind suggestions. We have revised the text (page number 11, last paragraph)

Round 2
Reviewer 2 Report
After reviewing the author response and the revised manuscript, although in some cases the author stated they implemented some modifications that they actually DID NOT, I would recommend this manuscript for publication after minor revision, upon REAL AMEDMENT of the reviewer suggestion.
Additionally, I would like to recommend the author to read (and in their case, re-red) the manuscript before sending it, at least to be aware of what they are actually sending.
Please, find below my comments to the new version:
In this manuscript, the authors reported the extraction of cellulose micr/nano fibres from Grewia optiva and used it for the encapsulation of C. roseus roots extract to prepare potential wound dressing. I strongly find this investigation of high interest, but I have some major concerns that made me recommend “Major revisions”.
· Abstract: In this section it is not clear the aim of the study. I mean which are the benefits of those bioactive extracts, which are those extract and their target. That should be briefly incorporated in this section. Indeed, although the general idea of what is done in this resarch can be easily seen, its novelty seems not to have been suitably highlighted.
Our response: Thank you very much kind suggestion to enhance the quality of our manuscript. We have elaborated the abstract and mentioned the novelty of our present work. (last sentences of abstract page)
Ok.
o Line 14: “Gravimetric analysis”…..In addition, the thermal character of the TGA result should be also highlighted, instead oof just “Gravimetric
Our response: Thank you very much for your kind suggestions. Herein we have performed the gravimetric analysis to quantify the component of raw fiber. We agree with reviewer that the TGA result would be more advantageous. However, due to the problem with our TGA instrument, we are unable to include the TGA graph. We will include the TGA data for upcoming manuscript. We hope you will consider situations.
Ok.
· Introduction:
o The authors used a changing referenc format (lines 51 and 55 for exampple). Please, unify it in the whole manuscript.
Our response: Thank you very much for minute correction. We have corrected the citation formatting throughout the text in revised manuscript (e.g page number 2, reference number 3).
Ok.
· Materials and Methods.
o Line 107: “…Gulmi district of Nepal in January 2019.” I would recommend the authors to extent a little bit origin of the sample, since that seems to be quite general.
Our response: Thank you very much for your kind suggestion. The description has been elaborated regarding the plant location in revised manuscript. (Page number 3, materials and methods part, and first paragraph)
Ok.
o Lines 117-120: Are the removal of lignin and hemicellulose by means of their solubilization and a subsequent filtration process? In that case (and if not as well) please complete this methodological part.
Our response: Thank you very much for your kind suggestion. We have added the following text in the e methodological part of revised manuscript (page number 3).
The dried samples were chopped and boiled in a toluene/ethanol mixture in the ratio of 2:1 in a Soxhlet to removed wax for 6 hour and dried for 24h in oven (70oC) to obtain nonwoven fibers. Later, the dried dewaxed fibers were bleached using 0.7% acidified NaClO2 in a magnetic stirrer at 90-95 ËšC for 3 h to remove of lignin content from the fiber. The hemicelluloses content of the sample was removed by treating with NaOH (4% w/v) at ~60 ËšC for 24 h in a magnetic stirrer. The resulting fibers were washed with distilled water and added to the mixture 250 mL of 4% NaOH and 3% H2O2 kept in a water bath at 80 ËšC for an hour followed by filtration. The obtained fibrous mass was added to 1% H2O2 (250 mL) and stirred for an hour at 85ËšC and washed with distilled water until the pH turned neutral and was allowed to air dry and weighed.
All this paragraph was lready included in the first version of the manuscript. It still remains unclear the removal of hemicellulose and lignin. Please clarify it and complete it. What does happen to the lignin and hemicelulose content when addding NaClO2 and NaOH respectively?
o Lines 140-144: Which were tha sample shapes?
Our response: Thank you very much for your kind concern. The samples were of 10 mm in diameter. We have added in revised manuscript (page number 9, antimicrobial test chapter)
Ok, but I would recommend to briefly indicate it also in the experimental section.
o Lines 170-177: Why did the authors select 10 min as the maximum water uptake time? Why not longer or shorter? Is the potential final product only applicable for that time span in an animal body?
Our response: Thank you very much for your kind concern. The water uptake efficiency of as-synthesized material remained constant after 10 min. Therefore we have showed the data for 10 min.
In that case, I consider it not appropriate since it does not represent the on-service conditions of the synthesized systems. Do the authors consider the information extracted from these results enough to extrapolate to longer lifespan in the animal bodies? This issue should be mentioned in the manuscript and somehow supported by other studies/investigation/results…
o Lines 188-203: All these lines seem to belong to the experimental section. I would recommend to summarize this part and complete that one located h¡in the experimental section.
Our response: Thank you very much for your kind suggestion. We have moved this part in experimental section in revised section.
These lines (corresponding to the lines 194-209 in the new version) are still in the same location…
o Line 221: It seems there is one missing reference.
Our response: Thank you very much for your minute correction. We have added the reference in revised manuscript.
Ok, but I would recommend to briefly indicate it also in the experimental section.
o Why did not the authors evaluate the porosity of these systems? That appears to be fundamental when dealing with these kind of materials and structures (even more importantly when considering their ultimate application target).
Our response: Dear reviewer, porosity has been calculated by apparent density (page number 6, last paragraph)
Dear authors, this parameter has been incorporated in the new version of the manuscript… I would recommend tocarefully read your own work before sending it for publication.
o The figure plotting the FTIR spectra is required.
Our response: Thank you very much for your kind concern. FTIR plot has been in revised manuscript.
I strongly recommend the authors to revise what they are actually sending to this journal, because after revising the manuscript, it only contains EXACTLY THE SAME FTIR FIGURE as in the previous version. Please include in the FTIR figure the lacking spectra…
o Line 293: “…(highlighted in green)” There is nothing highlighted in green. Please check it and rephrase it.
Our response: Thank you very much for your minute correction. We have corrected accordingly in revised manuscript.
Ok.
o Lines 185-299: The authors are encouraged to include a title (3.X…) of the first result section discussion.
Our response: Thank you very much for minute correction. We have revised accordingly ( 3.1 Physio-chemical characterization of G. optiva fiber, page number 5, 3.2 XRD, FTIR page number 7, 3.3 microscopic characterisation page number 8)
Ok.
o In figure 4 it is required to indicate which of these diagramas correspond to either CMF or CMF-E.
Our response: Thank you very much for your kind suggestions. We have revised Fig. 4 accordingly and included in revised manuscript page number 9)
Ok.
o Lines 351-352: “…extraction of C. roseus encapsulated CNF membrane can be a good choice for faster wound healing” Have the authors compared the velocity wound healing of their design dressing wound materials with some references? If not, this statement remains unfounded.
Our response: Thank you very much for your kind suggestions. We have revised the text (page number 11, last paragraph)
Author Response
Dear Reviewer,
Thank you very much for the kind inputs and feedbacks. We highly appreciate your efforts to make this manuscript well. We have made significant corrections in this draft. Please, find the edited manuscript with following major changes highlighted in a blue.
Please, find below my comments to the new version:
In this manuscript, the authors reported the extraction of cellulose micr/nano fibres from Grewia optiva and used it for the encapsulation of C. roseus roots extract to prepare potential wound dressing. I strongly find this investigation of high interest, but I have some major concerns that made me recommend “Major revisions”.
- Abstract: In this section it is not clear the aim of the study. I mean which are the benefits of those bioactive extracts, which are those extract and their target. That should be briefly incorporated in this section. Indeed, although the general idea of what is done in this resarch can be easily seen, its novelty seems not to have been suitably highlighted.
Our response: Thank you very much kind suggestion to enhance the quality of our manuscript. We have elaborated the abstract and mentioned the novelty of our present work. (last sentences of abstract page)
Ok.
o Line 14: “Gravimetric analysis”…..In addition, the thermal character of the TGA result should be also highlighted, instead oof just “Gravimetric
Our response: Thank you very much for your kind suggestions. Herein we have performed the gravimetric analysis to quantify the component of raw fiber. We agree with reviewer that the TGA result would be more advantageous. However, due to the problem with our TGA instrument, we are unable to include the TGA graph. We will include the TGA data for upcoming manuscript. We hope you will consider situations.
Ok.
- Introduction:
o The authors used a changing referenc format (lines 51 and 55 for exampple). Please, unify it in the whole manuscript.
Our response: Thank you very much for minute correction. We have corrected the citation formatting throughout the text in revised manuscript (e.g page number 2, reference number 3).
Ok.
- Materials and Methods.
o Line 107: “…Gulmi district of Nepal in January 2019.” I would recommend the authors to extent a little bit origin of the sample, since that seems to be quite general.
Our response: Thank you very much for your kind suggestion. The description has been elaborated regarding the plant location in revised manuscript. (Page number 3, materials and methods part, and first paragraph)
Ok.
o Lines 117-120: Are the removal of lignin and hemicellulose by means of their solubilization and a subsequent filtration process? In that case (and if not as well) please complete this methodological part.
Our response: Thank you very much for your kind suggestion. We have added the following text in the e methodological part of revised manuscript (page number 3).
The wax content from natural fiber is usually removed by boiling in polar solvent mixture such as benzene -ethanol azeotrope, toluene/ethanol mixture system [21]. Herein, we used the toluene/ethanol mixture system (2:1) to avoid the benzene toxicity [20]. Toluene dissolves wax and maintains fluidity at low temperatures while ethanol dissolves wax as well as many other polar compounds [22]. The dried samples were chopped and boiled in a toluene/ethanol mixture in the ratio of 2:1 in a Soxhlet to removed wax for 6 hour and dried for 24h in oven (70oC) and weighed in order to determine the content of the wax in the fiber. Later, the dried dewaxed fibers were bleached using 0.7% acidified NaClO2 in a magnetic stirrer at 90-95 ËšC for 3 h to remove of lignin content from the fiber. During this process the chemical bonds present in the chromophore were broken into small molecules that do not absorb visible light. The hemicelluloses content of the sample was removed by treating with NaOH (4% w/v) at ~60 ËšC for 24 h in a magnetic stirrer. Alkaline hydrolysis resulted the substantial breakdown of the lignocelluloses/ hemicelluloses structure and depolymerization of lignin by affecting the acetyl group in hemicelluloses and linkages of lignin-carbohydrate ester [23]. The resulting fibers were washed with distilled water and added to the mixture 250 mL of 4% NaOH and 3% H2O2 kept in a water bath at 80 ËšC for an hour followed by the filtration. The obtained fiberous mass was added to 1% H2O2 (250 mL) and stirred for an hour at 85ËšC and washed with distilled water until the pH turned neutral and was allowed to air dry and weighed
All this paragraph was already included in the first version of the manuscript. It still remains unclear the removal of hemicellulose and lignin. Please clarify it and complete it. What does happen to the lignin and hemicelulose content when addding NaClO2 and NaOH respectively?
Our response: Thank you for kind concern. We have elaborated and mentioned in experimental section.
Dear Editor, explanation for this treatment has included in the result and discussion session. As per your suggestion Lines 188-203: All these lines seem to belong to the experimental section. I would recommend to summarize this part and complete that one located h¡in the experimental section, we have edited this paragraph and included in the experimental section. We hope it will make a sense. Thank you.
o Lines 140-144: Which were tha sample shapes?
Our response: Thank you very much for your kind concern. The samples were of 10 mm in diameter. We have added in revised manuscript (page number 9, antimicrobial test chapter)
Ok, but I would recommend to briefly indicate it also in the experimental section.
Our response: Thank you for your suggestions. This sentence has been included into the experimental section of antimicrobial test.
Sample has been prepared by cutting the membrane into the small pieces with a diameter of 10mm
o Lines 170-177: Why did the authors select 10 min as the maximum water uptake time? Why not longer or shorter? Is the potential final product only applicable for that time span in an animal body?
Our response: Thank you very much for your kind concern. The water uptake efficiency of as-synthesized material remained constant after 10 min. Therefore we have showed the data for 10 min.
In that case, I consider it not appropriate since it does not represent the on-service conditions of the synthesized systems. Do the authors consider the information extracted from these results enough to extrapolate to longer lifespan in the animal bodies? This issue should be mentioned in the manuscript and somehow supported by other studies/investigation/results…
Our response: Dear Reviewer, thank you very much for very genuine and sensitive feedback. We have carefully reviewed our work and literature too. Careful examination of the literature has suggested that water uptake time of cellulose based microfiber has longer than hours…some paper has the data calculated in the hour. However, in this work initial measurement has been done in the interval of minute which after few minute seems constant but it might be the temporary retardation of uptake but not the optimum uptake condition. We appreciate your comment and would suggest the further research to carry out the experiment for hours not in minute.
We have added new references and elaborated in revised text
o Lines 188-203: All these lines seem to belong to the experimental section. I would recommend to summarize this part and complete that one located h¡in the experimental section.
Our response: Thank you very much for your kind suggestion. We have moved this part in experimental section in revised section.
These lines (corresponding to the lines 194-209 in the new version) are still in the same location…
Our response: Thank you for your kind suggestion. We have revised the experimental section incorporating these lines. These lines were shifted to the experimental section. Thank you.
o Line 221: It seems there is one missing reference.
Our response: Thank you very much for your minute correction. We have added the reference in revised manuscript.
Ok, but I would recommend to briefly indicate it also in the experimental section.
Dear reviewer, following paragraph has been added to the experimental section.
The chemical composition of GOFs fiber was measured by the gravimetric method. The method involved the chemical treatment followed by weight measurement and calculating weight loss.
o Why did not the authors evaluate the porosity of these systems? That appears to be fundamental when dealing with these kind of materials and structures (even more importantly when considering their ultimate application target).
Our response: Thank you for your kind concern. These parameters are crucial for cell infiltration and growth. The porosity has been calculated by apparent density (page number 6, last paragraph)
Dear authors, this parameter has been incorporated in the new version of the manuscript… I would recommend to carefully read your own work before sending it for publication.
Oure response: Thank you very much for your kind suggestion to improve the quality of our manuscript. We have revised accordingly.
o The figure plotting the FTIR spectra is required.
Our response: Thank you very much for your kind concern. FTIR plot has been in revised manuscript.
I strongly recommend the authors to revise what they are actually sending to this journal, because after revising the manuscript, it only contains EXACTLY THE SAME FTIR FIGURE as in the previous version. Please include in the FTIR figure the lacking spectra…
Oure response: Thank you very much for your valuable suggestion. We applies for our mistake. We have revised the FTIR spectra in revised manuscript and indicated the peak of interest in revised figure. We hope it is more clear now. In addition, we separated the XRD and FTIR section.
o Line 293: “…(highlighted in green)” There is nothing highlighted in green. Please check it and rephrase it.
Our response: Thank you very much for your minute correction. We have corrected accordingly in revised manuscript.
Ok.
o Lines 185-299: The authors are encouraged to include a title (3.X…) of the first result section discussion.
Our response: Thank you very much for minute correction. We have revised accordingly ( 3.1 Physio-chemical characterization of G. optiva fiber, page number 5, 3.2 XRD, FTIR page number 7, 3.3 microscopic characterisation page number 8)
Ok.
o In figure 4 it is required to indicate which of these diagramas correspond to either CMF or CMF-E.
Our response: Thank you very much for your kind suggestions. We have revised Fig. 4 accordingly and included in revised manuscript page number 9)
Ok.
o Lines 351-352: “…extraction of C. roseus encapsulated CNF membrane can be a good choice for faster wound healing” Have the authors compared the velocity wound healing of their design dressing wound materials with some references? If not, this statement remains unfounded.
Our response: Thank you very much for your kind suggestions. We have revised the text (page number 11, last paragraph)